# Bi-terminal fusion of intrinsically-disordered mussel foot protein fragments boosts mechanical strength for protein fibers

Jingyao Li[1], Bojing Jiang [1], Xinyuan Chang[1], Han Yu[1], Yichao Han [1] & Fuzhong Zhang [1,2,3] ✉

Microbially-synthesized protein-based materials are attractive replacements for petroleum-derived synthetic polymers. However, the high molecular weight, high repetitiveness, and highly-biased amino acid composition of high-performance protein-based materials have restricted their production and widespread use. Here we present a general strategy for enhancing both strength and toughness of low-molecular-weight protein-based materials by fusing intrinsically-disordered mussel foot protein fragments to their termini, thereby promoting end-to-end protein-protein interactions. We demonstrate that fibers of a ~60 kDa bi-terminally fused amyloid-silk protein exhibit ultimate tensile strength up to $481 \pm 31$ MPa and toughness of $179 \pm 39$ MJ*m$^{-3}$, while achieving a high titer of $8.0 \pm 0.70$ g/L by bioreactor production. We show that bi-terminal fusion of Mfp5 fragments significantly enhances the alignment of β-nanocrystals, and intermolecular interactions are promoted by cation-π and π-π interactions between terminal fragments. Our approach highlights the advantage of self-interacting intrinsically-disordered proteins in enhancing material mechanical properties and can be applied to a wide range of protein-based materials.

Microbially-synthesized materials are attractive replacements for petroleum-derived synthetic polymers as they are renewable and biodegradable. Recent advances in synthetic biology have offered the possibility to produce protein-based materials (PBMs) with precisely controlled protein sequences, functional domains, and uniform molecular weights, leading to high yet tunable mechanical performances beyond the reach of natural protein materials and petroleum-derived synthetic polymers[1–3]. One example is spider silk, best known for its unique combination of high tensile strength and high toughness[4,5]. This attractive mechanical property has motivated decades of research to produce silk and silk-mimetic materials from fast-growing microbial hosts[6–8]. Most high mechanical-performing protein-based materials have high molecular weight (MW), repetitive protein sequences, and highly biased amino acid compositions[7,9–12]. These

sequence features are essential to the mechanical properties of PBMs but have presented major technical obstacles for their biosynthesis in microbial hosts, such as limited choices of amino acids and genetic instability. Current approaches to solve these problems include post-translational protein polymerization[12,13], host engineering[9], and engineering biomimetic fiber spinning[14]. Through these efforts, recombinant spider silk protein and engineered silk-amyloid hybrid protein containing 192 and 128 repeating units with MW of 556 kDa and 378 kDa were synthesized from microbial hosts, respectively[10,12]. Fibers made of these high MW PBMs displayed high mechanical properties (tensile strength up to ~1 GPa, toughness up to ~160 MJ*m$^{-3}$), similar to or higher than those of natural silk fibers[10]. Despite this progress, producing high MW PBMs with high mechanical performances in microbial hosts often leads to low titers and yields. Metabolic and

[1]Department of Energy, Environmental and Chemical Engineering, Washington University in St. Louis, One Brookings Drive, Saint Louis, MO 63130, USA. [2]Division of Biological & Biomedical Sciences, Washington University in St. Louis, One Brookings Drive, Saint Louis, MO 63130, USA. [3]Institute of Materials Science & Engineering, Washington University in St. Louis, One Brookings Drive, Saint Louis, MO 63130, USA. ✉e-mail: fzhang@seas.wustl.edu

genetic engineering methods are useful in enhancing protein titers and yields[15]; however, these strategies have limited effects on high MW PBMs[9,12]. Meanwhile, a less explored strategy is to use protein engineering to achieve high mechanical performance from low MW proteins that can be produced in high titers and yields from microbial hosts.

Mussel foot protein 5 (Mfp5) is an intrinsically disordered protein secreted at the tip of mussel byssus and is used for surface adhesion underwater[16,17]. Mfp5 contains multiple tyrosine residues that can be enzymatically oxidized to L-3,4-dihydroxyphenylalanine (DOPA) and play important roles in surface interacting. Beside surface adhesion, the disordered and flexible Mfp5 also self-interacts under neutral and basic conditions through π-π interaction, cation-π interaction, and hydrophobic effects, thus playing important cohesive roles[18]. Moreover, recent experimental and modeling results have shown that mussel foot proteins remain adhesive and cohesive even without posttranslational modifications[19,20].

Inspired by the molecular glue Mfp5, here we demonstrate a general approach to promote both the strength and the toughness of low MW protein fibers by genetically fusing Mfp5 fragments to their termini, therefore promoting end-to-end protein-protein interactions (Fig. 1). We applied this strategy to multiple proteins, including recombinant silk repeats, amyloid-silk hybrid proteins, a titin fragment, Fyn tyrosine kinase SH3 domain (SH3), and green fluorescent protein (GFP). Our results showed that in all these cases, bi-terminal Mfp5 (btMfp5) fusion promoted fiber ultimate tensile strength (by up to 345%) and toughness (by up to 1,970%). Using site-directed mutagenesis, we further showed that the significantly enhanced fiber mechanical properties are mostly contributed by interactions of N-half tyrosine with C-half tyrosine and/or cationic residues. One of the btMfp5-fused amyloid-silk polymers, [N]M-16xFGA-[C]M(YtoS), has an MW of 57.3 kDa, but its fiber displayed an ultimate tensile strength of 481 ± 31 MPa and a toughness of 179 ± 39 MJ*m$^{-3}$. These properties are comparable to recombinant spider silk of 285 kDa. This low MW protein has enabled high titer/yield protein production, resulting in a titer of 8.0 ± 0.70 g/L and a yield of 13.5 ± 1.0% of the total proteome. Our protein engineering strategy together with proteins designed in this project will enable a broad

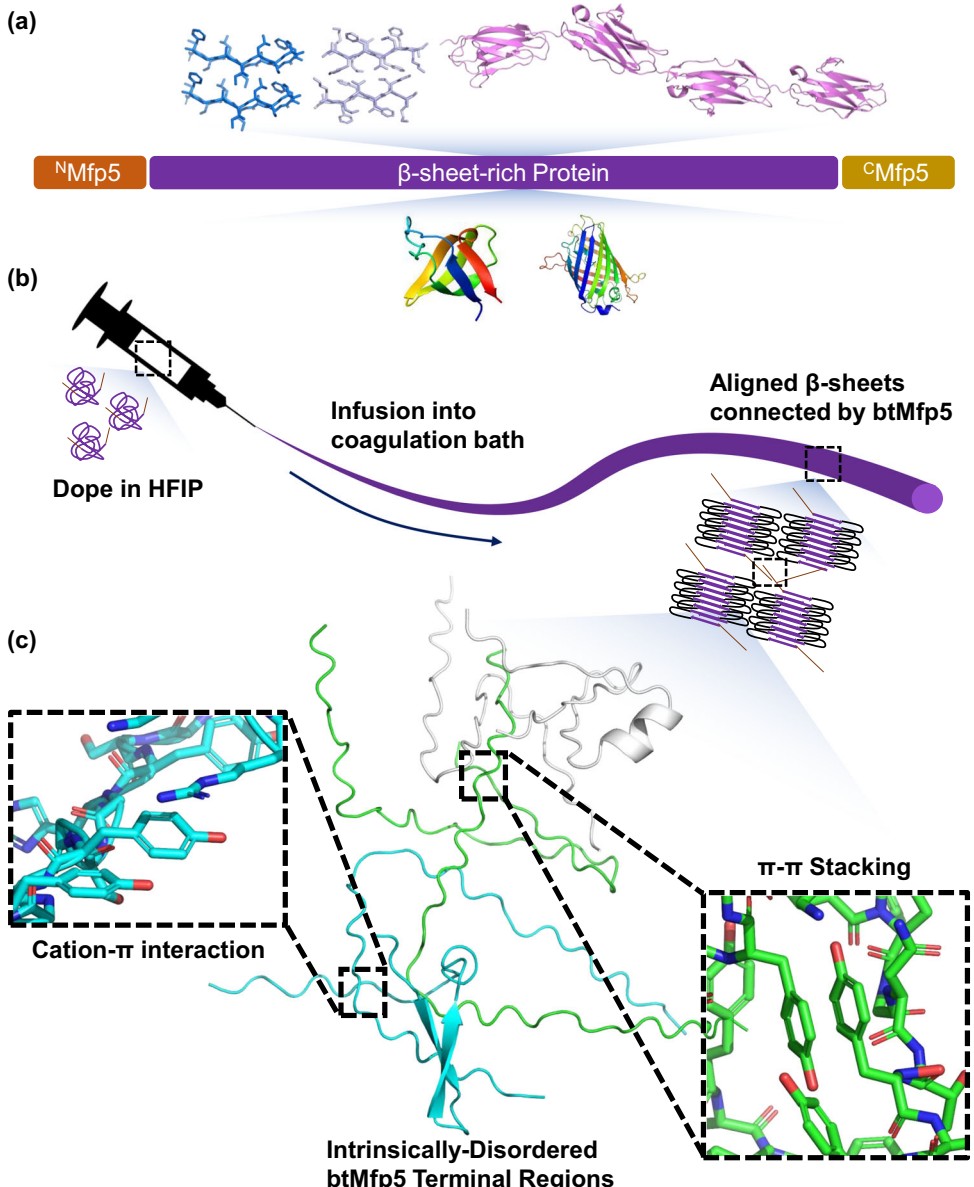

**Fig. 1 | Schematics of the btMfp5-fused protein fiber spinning. a** Schematics of recombinant proteins fused with split Mfp5 fragments at their termini. **b** Schematics of protein structural change during fiber spinning. **c** Schematics of interactions between split Mfp5 fragments. Only Mfp5 fragments are shown.

range of material applications in the fields of protection, sustainability, and biomedical technology[21,22].

## Results

### Bi-terminal fusion of Mfp5 fragments to amyloid-silk protein 16xFGA promotes its fiber mechanical properties

Fiber wet-spinning requires the preparation of highly concentrated protein dope and the protrusion of the dope into a coagulation bath that promotes protein interaction. To avoid loss-of-function during fiber spinning, we focused on IDPs that can strongly interact in disordered forms while remain soluble for dope preparation. One such candidate is Mfp5, whose self-interaction can be controlled by pH and solvents. At acidic pH or the presence of disrupting solvents (e.g. acetic acid or 1,1,1,3,3,3-hexafluoroisopropanol, HFIP), Mfp5 is soluble due to its high positive charge and favorable protein-solvent interactions. When placed in neutral or basic aqueous solutions, Mfp5 self-interacts by forming multiple cation-π and π-π interactions between positively charged residues and tyrosine/DOPA sidechain[20,23]. Such self-interaction can form even in the absence of DOPA residues, thus does not require additional enzymatic oxidation in protein preparation. The full-length Mfp5 has a MW of 8.9 kDa and contains 74 amino acids, including 20 tyrosine and 16 lysine residues. We split Mfp5 in the middle into two halves with similar number of residues and MW on each segment (Fig. 1a). Upon splitting, the N-half (NMfp5, NM) has 10 Ys but only 3 Ks and has a net charge of +1 at pH = 7, while the C-half (CMfp5, CM) contains 10 Ys, 13 Ks, and 2 Rs with a net charge of +15 at pH = 7 in aqueous solution (Supplementary Table 1, Supplementary Dataset 1).

We first fused the split Mfp5 fragments to the termini of a 16-repeat of artificially-designed amyloid-silk protein 16xFGA (Fig. 2a)[10]. The resulting fusion protein, namely NM-16xFGA-CM, was expressed and purified as a soluble protein in denaturing or acidic aqueous solutions (Supplementary Figure 1). NM-16xFGA-CM is also highly soluble in HFIP (up to 30% w/v) as HFIP acts as a good H-bond donor, thus a concentrated protein dope can be readily prepared and remains stable (without precipitation or aggregation). Protein dope was protruded into an 95% methanol bath, which effectively removed HFIP and promoted non-covalent interactions between Mfp5 fragments (Fig. 1)[24,25]. Fibers produced from this procedure were compared to fibers made of the 16xFGA protein[10]. Optical microscopy and scanning electron microscopy (SEM) revealed that the NM-16xFGA-CM fibers had regular cylindrical shapes and consistent diameters of $19 \pm 0.70 \,\mu m$ (Fig. 2b, c). Standard tensile tests showed that btMfp5 fusion has enhanced both fiber strength and toughness. The ultimate tensile strength increased by 77% from $230 \pm 34$ MPa of 16xFGA fibers[10] to $406 \pm 36$ MPa of NM-16xFGA-CM fibers, and toughness increased by 100% from $59 \pm 17$ MJ*m$^{-3}$ of 16xFGA fibers to $118 \pm 21$ MJ*m$^{-3}$ of NM-16xFGA-CM fibers (Fig. 2d–g, Supplementary Figure 2, Supplementary Table 2)[10]. Mechanical enhancement on recombinant silk fibers was observed when globular terminal domains of spidroins were fused to a minimal repetitive domain (2 repeats)[14,26]. Here we have shown that bi-terminal fusion of the intrinsically disordered self-interacting Mfp5 fragments, which have significantly different structure and sequence than silk terminal domains, can also enhance fiber mechanical properties. Our result thus highlighted the potential of engineering strong fiber materials from the vast protein sequence space.

### btMfp5 fusion promotes the alignment of β-sheets along fiber axis

After observing the enhanced mechanical properties from btMfp5 fusion, we aimed to investigate the underlying mechanism and structural influence on fibers. Except for oligomerization to reduce breaks at chain ends, terminal domains of natural spidroins also involve in fiber assembly by triggering β-sheet formation and their alignment along the fiber axis[27–30]. In natural silk fibers, the alignment of β-sheets to the fiber axis is an important factor contributing to the exceptional

mechanical properties[7,31,32]. Our previous work has shown that the 16xFGA amyloid-silk repeats can trigger β-nanocrystal formation[10], and here we hypothesized that inter-molecular interactions between bi-terminal fused Mfp5 fragments would facilitate the alignment of β-sheet along fiber axis. To test this hypothesis, we used polarized Raman spectrometry to examine the amide I β-sheet peak (1670 cm$^{-1}$) and compared the Raman spectra when fibers was oriented either parallel (ZZ) and perpendicular (XX) to the direction of laser polarization[12,28,33,34]. For 16xFGA fibers, their amide I β-sheet peak measured at perpendicular position ($I_{XX}$) is only slightly stronger than that at parallel position ($I_{ZZ}$). The peak intensity ratio between perpendicular and parallel positions $I_{Y/X}$ was $1.13 \pm 0.04$, suggesting that β-sheets in 16xFGA fibers are only weakly aligned to the fiber axes. In contrast, amide I β-sheet peak of NM-16xFGA-CM fibers are dramatically higher at the perpendicular position ($I_{XX}$) than at the parallel position ($I_{ZZ}$), indicating a substantially enhanced β-sheet alignment compared to 16xFGA fibers (Fig. 2j). The peak intensity ratio reached $1.68 \pm 0.06$, significantly higher than that of previously reported high MW (556 kDa) recombinant spider silk[12].

### Tyrosine and positively charged residues form end-to-end intermolecular interactions between Mfp5 fragments

Upon observing the enhanced mechanical properties from btMfp5 fusion, we aimed to understand the interaction mechanism. We first fused the full-length Mfp5 to the C-terminus of 16xFGA. The resulting fusion protein 16xFGA-M has the same amino acid composition and MW with NM-16xFGA-CM, but its Mfp5 can only interact with other protein molecules from one end (Supplementary Figure 3). We found that 16xFGA-M fibers failed to exhibit comparable ultimate tensile strength with NM-16xFGA-CM fibers (Supplementary Table 3), indicating that it is necessary for the Mfp5 fragments to be fused to both protein termini to promote end-to-end intermolecular interactions. Next, we examined which residues are involved in the end-to-end interactions. The dominant form of interaction between Mfp5 are cation-π interaction (Y-K/R)[20,35,36] and π-π stacking (Y-Y)[37]. Previous research mostly emphasized the former and estimated a binding energy of −20.8 kcal/mol per Y-K pair[38]. More recently, ab initio quantum mechanical simulation suggested that total binding energy from π-π stacking can add up to 33-50% of the cation-π interactions[36]. To differentiate these interactions, we designed, biosynthesized, and purified a series of NM-16xFGA-CM mutant proteins (Fig. 3a–c). The N-half Mfp5 (NM) is most enriched with Y with only 3 positively charged residues at neutral pH. We thus mutated all Ys to S, resulting in NM(YtoS)−16xFGA-CM (Fig. 3a, Supplementary Figure 4–6). The ultimate tensile strength of NM(YtoS)−16xFGA-CM fibers was close to those of 16xFGA fibers but was significantly lower than those of NM-16xFGA-CM fibers (Fig. 3d–f, Supplementary Figure 4–6, Supplementary Table 4), indicating that tyrosine residues in NM play vital roles in the end-to-end Mfp5 interaction. Interestingly, NM(YtoS)−16xFGA-CM fibers increased fiber-breaking strain compared to 16xFGA fibers, leading to an increase in fiber toughness. This suggests that although YtoS mutant cannot hold tensile stress beyond 260 MPa, the flexible NM(YtoS) mutant allows protein chains to be straightened and weakly-interact under extension. Additionally, the C-half Mfp5 (CM) has 10 tyrosine and 15 positively charged residues (13 Ks and 2 Rs). We separately mutated all Ys and positively charged residues to S, resulting in two CM mutants, NM-16xFGA-CM(YtoS) and NM-16xFGA-CM(KRtoS), respectively (Fig. 3b, Supplementary Figure 4–6). Fibers of both CM mutants exhibited similar mechanical properties with NM-16xFGA-CM fibers (Fig. 3d–f, Supplementary Figure 4–6, Supplementary Table 5–6), suggesting that either cation-π interaction between C-half positively-charges residues and N-half Ys or π-π stacking between Ys in both halves are sufficient to enhance fiber mechanical properties. The ultimate tensile strength and toughness of NM-16xFGA-CM(KRtoS) fibers reached $424 \pm 58$ MPa and $130 \pm 14$ MJ*m$^{-3}$.

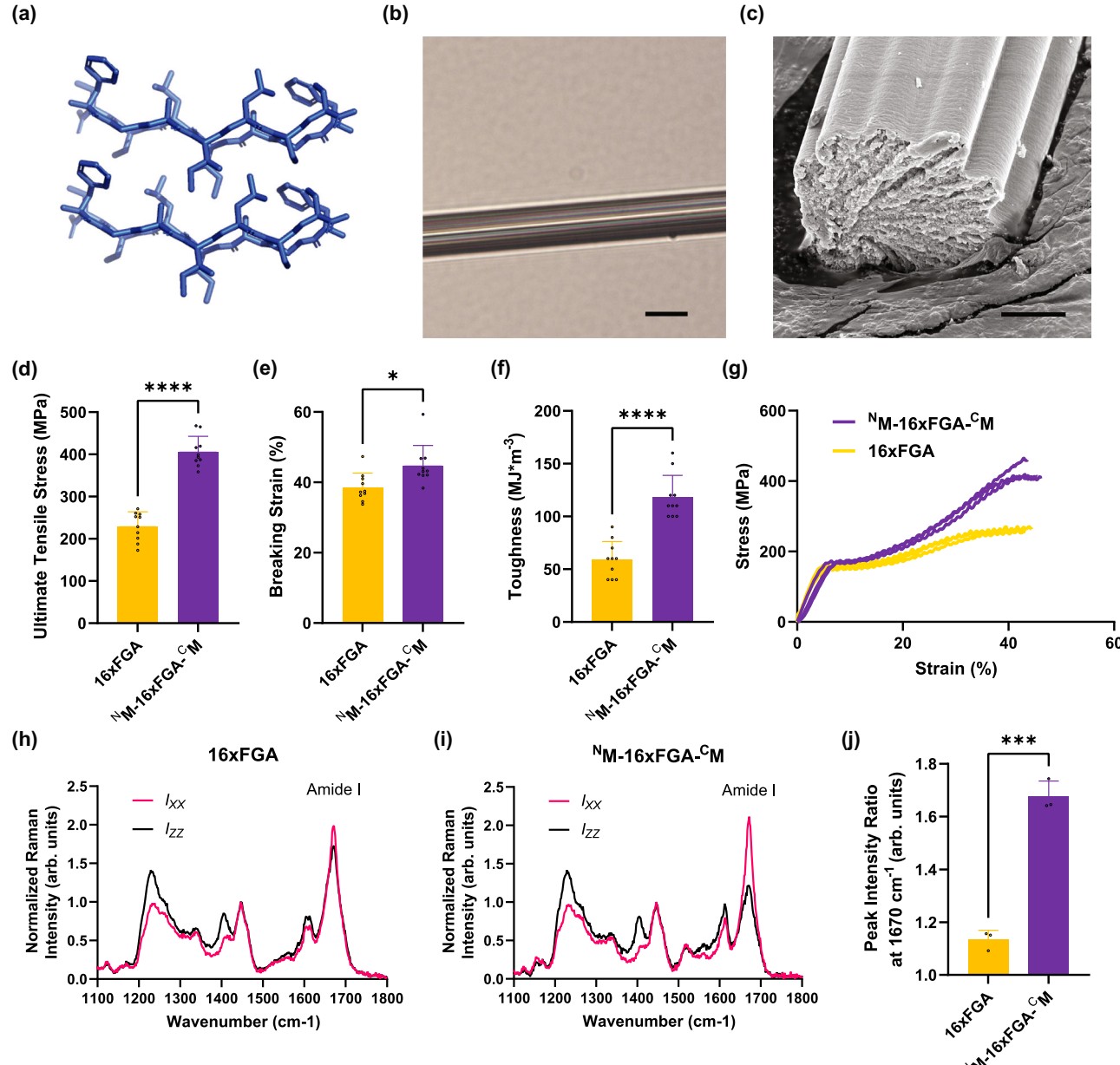

**Fig. 2 | Mechanical properties and polarized Raman spectra of 16xFGA and NM-16xFGA-CM fibers. a** Crystal structures of self-assembled FGAILSS (PDB ID: 5E61) in the cross-β form. **b** Optical microscopy image of NM-16xFGA-CM fiber. The scale bar in the image was 20 μm. **c** SEM image of NM-16xFGA-CM fiber. The scale bar in the image was 5 μm. **d**–**f** Ultimate tensile strength (**d**, ****$p = 1.6 \times 10^{-9}$), breaking strain (**e**, *$p = 1.1 \times 10^{-2}$) and toughness (**f**, ****$p = 2.0 \times 10^{-6}$) of 16xFGA and NM-16xFGA-CM fibers, two-tailed unpaired t-test. **g** Representative stress-strain curves of 16xFGA and NM-16xFGA-CM fibers. **h**–**j** Polarized Raman spectra of 16xFGA (**h**) and NM-16xFGA-CM fibers (**i**), and peak intensity ratio at 1670 cm⁻¹ of the two fibers

(**j**, ***$p = 1.6 \times 10^{-4}$), two-tailed unpaired t-test. Data are presented as mean values ± standard deviations. Error bars represent standard deviation. 16xFGA mechanical data reproduced from previous publication (reproduced with permission[10], Copyright 2021 American Chemical Society) serve as a comparison. For all mechanical data, $n = 10$ independent tensile test measurements; for peak intensity ratio, $n = 3$ ratios of a pair of peak intensity at 1670 cm⁻¹ from independently acquired Raman spectra on the $I_{XX}$ and the $I_{ZZ}$ directions. Details of source data and statistical analyses are provided in the Source Data file.

To further confirm the roles of cation-π interaction, we examined fiber properties under different pH. As pH decreases, Mfp5 fragments are expected to be more strongly positively charged, thus promoting more cation-π interactions and further enhancing fiber mechanical properties. To test this hypothesis, we spun the NM-16xFGA-CM(YtoS) protein at acidic and basic pH from 5.5 to 11 (see Methods). We chose the NM-16xFGA-CM(YtoS) protein because its CM lacks tyrosine, thus reducing the influence from π-π interaction. Consistent with our hypothesis, ultimate tensile strength of fibers spun at pH 5.5 was 18% and 31% higher than those spun at pH 8.0 and pH 11, while toughness was 48% and 67% higher than those spun at pH 8.0 and pH 11,

respectively (Fig. 3g–i, Supplementary Fig. 7, Supplementary Tables 8–9). NM-16xFGA-CM(YtoS) fibers spun at pH 5.5 exhibited an impressive ultimate tensile strength of 481 ± 31 MPa and toughness of 179 ± 39 MJ*m⁻³. These results not only support the role of cation-π interactions between Mfp5 fragments as an important mechanism, but also suggest strategies in optimizing process conditions (e.g. spin in acidic coagulation bath) to promote fiber mechanical properties.

Additionally, because Mfp5 is intrinsically disordered, Mfp5 fragments in these fibers most likely do not fold into a specific structure when they interact. The end-to-end intermolecular interactions may occur between any positively charged residue

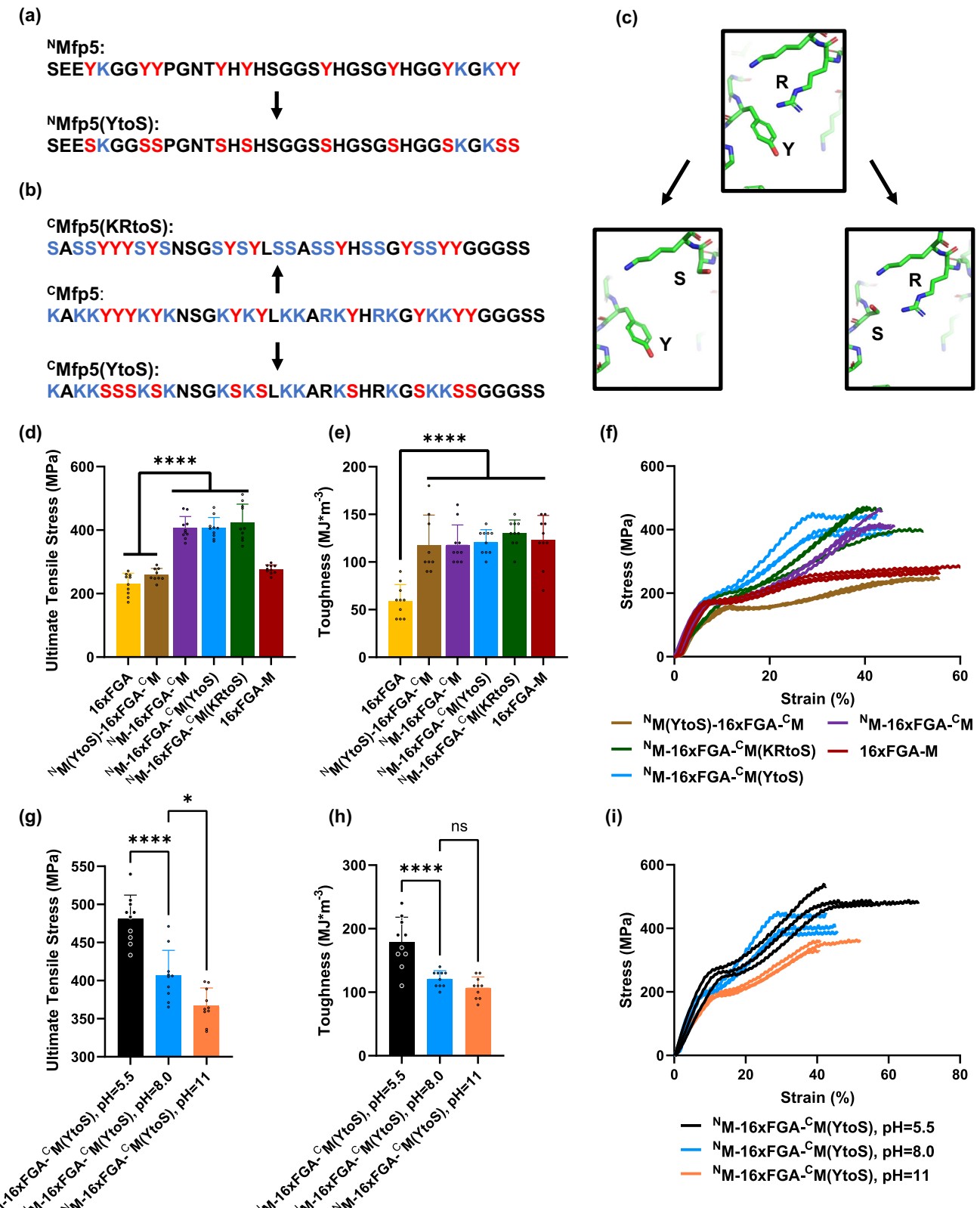

(a)

ᴺMfp5:
SEEYKGGYYPGNTYHYHSGGSYHGSGYHGGYKGKYY

ᴺMfp5(YtoS):
SEESKGGSSPGNTSHSHSGGSSHGSGSHGGSKGKSS

(b)

ᶜMfp5(KRtoS):
SASSYYYSYSNSGSYSYLSSASSYHSSGYSSYYGGGSS

ᶜMfp5:
KAKKYYYKYKNSGKYKYLKKARKYHRKGYKKYYGGGSS

ᶜMfp5(YtoS):
KAKKSSSKSKNSGKSKSLKKARKSHRKGSKKSSGGGSS

properties of ᶜM-16xFGA-ᶜM fibers are similar to those of ᴺM-16xFGA-ᶜM fibers, and significantly higher than those of 16xFGA fibers (Supplementary Fig. 9, Supplementary Table 7). These results indicate that interactions between disordered ᶜM and ᶜM can also promote fiber mechanical properties when fused to both protein termini.

and tyrosine from each bi-terminally-fused Mfp5 fragment. And such end-to-end intermolecular interactions are not limited to the current split Mfp5 fragments but can be potentially achieved by other IDPs that are rich in tyrosine and positively charged residues. To test so, we created another mutant ᶜM-16xFGA-ᶜM that has ᶜM on both termini (Supplementary Fig. 8). Mechanical

**Fig. 3 | Design and mechanical properties of btMfp5-fused 16xFGA variant fibers. a** Design of $^N$Mfp5(YtoS) sequence from $^N$Mfp5 with all tyrosine residues mutated to serine. Y and S mutated from Y were labeled in red, while K/R were labeled in blue. **b** Design of $^C$Mfp5(YtoS) and $^C$Mfp5(KRtoS) from $^C$Mfp5 with all tyrosine or lysine/arginine residues mutated to serine. Y and S mutated from Y were labeled in red, while K/R and S mutated from K/R were labeled in blue. **c** Schematics of mutations introduced to the split Mfp5 fragments. **d** Ultimate tensile strength of 16xFGA, 16xFGA-M, four btMfp5-fused 16xFGA variant fibers. **e** Toughness of 16xFGA, 16xFGA-M, four btMfp5-fused 16xFGA variant fibers. **f** Representative stress-strain curves of four btMfp5-fused 16xFGA variant fibers. **g** Summary of ultimate tensile stress for $^N$M-16xFGA-$^C$M(YtoS) spun in different pH in 95%

methanol; **h** summary of toughness for $^N$M-16xFGA-$^C$M(YtoS) spun in different pH in 95% methanol; **i** representative stress-strain curves of $^N$M-16xFGA-$^C$M(YtoS) spun in different pH in 95% methanol. Data are presented as mean values ± standard deviations. Error bars represent standard deviation. *$p < 0.05$, ****$p < 0.0001$, ordinary one-way ANOVA multiple comparisons. 16xFGA mechanical data reproduced from previous publication (reproduced with permission[10], Copyright 2021 American Chemical Society) serve as a comparison. For all mechanical data other than $^N$M(YtoS)−16xFGA-$^C$M fibers, $n = 10$ independent tensile test measurements; for $^N$M(YtoS)−16xFGA-$^C$M fibers, $n = 9$ independent tensile test measurements. Details of source data and statistical analyses are provided in the Source Data file.

## btMfp5 fusion promotes mechanical properties for a wide range of protein fibers

Upon demonstrating the enhanced mechanical property for 16xFGA fibers, we examined whether btMfp5 fusion can be a general method for promoting the mechanical properties of other protein fibers. We fused split Mfp5 fragments to three types of fiber proteins, including a recombinant spider silk 16xAAA (polyalanine)[9], another amyloid-silk hybrid protein 16xKLV[10], and a titin fragment Ig67-70[11]. These fiber materials use different molecular interaction mechanisms and form different sizes and portion of β-nanocrystals[10,11]. Bi-terminal fusion of split Mfp5 fragments enhanced fiber ultimate tensile strength by 65.5-139% for all three proteins (Fig. 4a–i, Supplementary Figs. 10–12, Supplementary Tables 10–13). Interestingly, we observed decreased breaking strain for btMfp5-fused 16xKLV and titin fibers, which may be a consequence of the interacting Mfp5 fragments hindering the free movement of peptide chains in the amorphous region, therefore limiting their capacity to extend before the load is transferred to the crystalline region. Polarized Raman spectrometry showed increased $I_{Y/X}$ peak intensity ratios for all three btMfp5-fused fibers, confirming enhanced β-sheet alignment along the fiber axis for all three types of protein materials (Supplementary Fig. 13). In the case of 16xAAA fibers, the degree of crystallinity was much lower as indicated by previous WAXS results[10], and the interaction among Mfp5 fragments was likely to have enhanced the chain entanglement in the amorphous region, which could explain the elevated breaking strain with btMfp5 fusion.

We further tested whether btMfp5 fusion could convert globular proteins that have never been used as fiber materials. We chose GFP which might endow fibers with fluorescence properties, and the SH3 domain, which is commonly used as a model globular protein for protein folding (Fig. 4m, q). Both proteins were not capable of forming strong intermolecular interaction. Therefore, they were not expected to yield strong fibers. Using the same wet-spinning technique as described above, we obtained GFP and SH3 fibers, and both fibers were indeed very weak and brittle (Fig. 4n, r). However, btMfp5 fusion ($^N$M-GFP-$^C$M and $^N$M-SH3-$^C$M) substantially enhanced both fibers' major mechanical properties. The strengths of GFP and SH3 fibers are enhanced by 345% and 54.9% to 249 MPa and 127 MPa, respectively. The breaking strains were increased by 11.6- and 3.43-folds, respectively. Most impressively, the toughnesses were enhanced by 19.7- and 19.2-fold to 59 and 100 MJ*m$^{-3}$, respectively (Fig. 4m–t, Supplementary Figure 14–16, Supplementary Table 14–17). The $^N$M-GFP-$^C$M fibers remained fluorescent under blue light excitation (Fig. 5a, Supplementary Figure 17). Overall, these results showed that btMfp5 fusion is a general and highly effective strategy to boost mechanical properties of protein fibers.

At last, to demonstrate the potential of btMfp5 fusion in facilitating scaled-up production, we expressed $^N$M-16xFGA-$^C$M in a 2 L fed-batch bioreactor (Supplementary Fig. 18). The final $OD_{600}$ of the culture reached 155. A high protein expression level of $13.5 ± 1.0\%$ was obtained, and the titer of the protein was determined as $8.0 ± 0.70$ g/L with the standard curve, which was generated with bands of purified btMfp5-fused 16xFGA proteins with known concentrations (Supplementary Figs. 19–21, Supplementary Note). This represents a five-to-ten-fold enhancement on the protein titer compared to that past

studies, such as a 285 kDa recombinant silk protein (containing 96 repeats)[39], while their fibers display comparable strength and toughness, falling well within the range of conventional natural and synthetic fibers (Fig. 5b, Supplementary Fig. 22 and Supplementary Table 18)[12].

## Discussions

Our results demonstrate a general approach to improve mechanical properties of protein fibers using btMfp5 fusion, allowing fibers made of relatively low MW proteins of 20-60 kDa to display high tensile strength and toughness (up to 481 MPa and 179 MJ/m³), which has only been observed previously on high MW recombinant silk proteins with MW >285 kDa. While these large proteins can eventually generate strong fibers, their yields were significantly lower than low MW proteins as demonstrated in this study[12,39]. Furthermore, high MW recombinant silk proteins often require complicated purification processes such as ammonia sulfate precipitation and multi-step chromatography (e.g. size exclusion chromatography)[9,10,12,40], while our btMfp5-fused low MW proteins can be purified using one-step affinity chromatography. Additionally, we show that the enhanced strength and toughness was caused by end-to-end intermolecular interactions between tyrosine in $^N$Mfp and tyrosine or positively charged residues in $^C$Mfp. This approach is effective on a wide range of proteins, including traditional fiber materials (i.e. silk and amyloid-silk) as well as globular proteins (i.e. GFP and SH3) that are not traditionally used as bulk materials, thereby creating fibers with interesting properties, such as fluorescence.

This approach is widely applicable because the fused Mfp5 fragments are disordered and the fiber spinning process is not affected by the unfolding of proteins. Previously, water-soluble terminal domains of natural spidroins were fused to two repeating units of recombinant silk protein[14,26]. During fiber spinning, these terminal domains undergo NTD-NTD and CTD-CTD dimerization in response to pH change and shearing force, thus allowing multiple molecules to form non-covalently-associates and increasing *de facto* MW. This strategy elegantly enabled aqueous spinning similar to that used by spiders[14,26,41–44]. Nevertheless, fusing the terminal domains to a large number of repeats or other proteins often leads to misfolding of the chimeric proteins, thus limiting its use[27,45]. Our method exploited the design principle of silk terminal domains, but used the disordered Mfp5, whose self-interactions were not affected by misfolding and could be controlled during fiber spinning. Additionally, disordered Mfp5 fragments have an estimated radius of gyration of 12–16 Å[46,47], allowing them to sample a large space for protein-protein interaction. Interaction between Mfp5 fragments can also adopt numerous configurations (e.g. between different residues) and orientations (Fig. 1c). Both advantageous features of IDP have greatly enhanced the chance of end-to-end protein interaction, thereby effectively reducing defects caused by chain-ends and promoting fiber mechanical properties. Although in solution, Mfp5 fragments within a chimeric protein can undergo intramolecular interactions if the N- and C-termini of the fused protein are in close proximity, the shear force during fiber spinning is expected to stretch protein chains along the fiber axis and effectively separate them, thus promoting the oligomerization of fused PBMs.

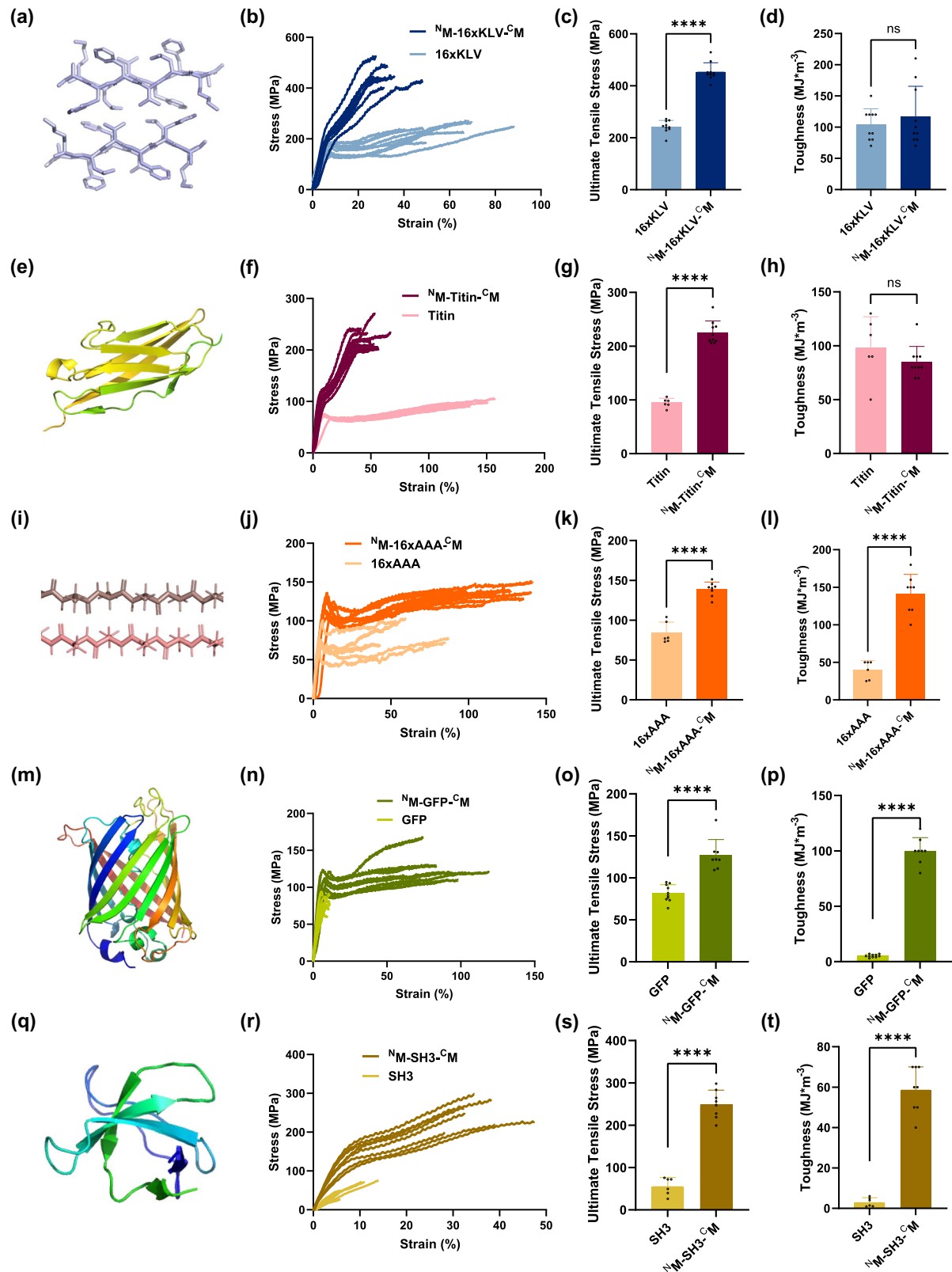

Tyrosine residues of natural Mfp5 are enzymatically oxidized to DOPA[48], which play key roles in protein-protein cohesive interaction and surface adhesion[23]. Adhesions to metal, metal oxide and silica surfaces are particularly strong due to the formation of high energy chelation and bi-dentate hydrogen bonds[23,49]. However, recent studies demonstrated that Mfp and Mfp-derived peptides without tyrosine-DOPA conversion showed a comparable level of cohesive-interaction and adhesion to surfaces such as mica or hydrophilic glass[19,20,37,50]. Experimental and molecular dynamic (MD) studies further revealed the strong interaction between tyrosine sidechains and positively charged residues in Mfp-Mfp interaction[19,20]. Tyrosine in Mfp5 fragments used in this study remained unmodified, thus largely

**Fig. 4 | Bi-terminal Mfp5 fusion boosted mechanical properties of fibers spun from other silk-forming/β-sheet-rich non-silk-forming proteins. a–d** Amyloid steric zipper structure (PDB ID: 3OW9), stress-strain curves, ultimate tensile stress and toughness of 16xKLV and $^N$M-16xKLV-$^C$M fibers; **e–h** representative Ig structure (PDB ID: 3B43), stress-strain curves, ultimate tensile stress and toughness of methanol-spun titin and $^N$M-titin-$^C$M fibers; **i–l** representative secondary structure element in β-nanocrystals in polyalanine spider silk, stress-strain curves, ultimate tensile stress and toughness of 16xAAA and $^N$M-16xAAA-$^C$M fibers; **m–p** structure (PDB ID: 1EMB), stress-strain curves, ultimate tensile stress and toughness of GFP and $^N$M-GFP-$^C$M fibers; **q–t** Structure (PDB ID: 3UA6), stress-strain curves, ultimate tensile stress and toughness of SH3 and $^N$M-SH3-$^C$M fibers. Data are presented as mean values ± standard deviations. Error bars represent standard deviation. ns $p \geq 0.05$, no significant difference; ****$p < 0.0001$, two-tailed unpaired $t$-test. 16xKLV and 16xAAA mechanical data reproduced from previous publication (reproduced with permission[10], copyright 2021 American Chemical Society) serve as comparisons. For 16xKLV, $^N$M-16xKLV-$^C$M, $^N$M-titin-$^C$M and GFP fibers, $n = 10$ independent tensile test measurements; for $^N$M-16xAAA-$^C$M, $^N$M-GFP-$^C$M fibers and $^N$M-SH3-$^C$M fibers, $n = 8$ independent tensile test measurements; for methanol-spun titin, 16xAAA and SH3 fibers, $n = 6$ independent tensile test measurements. Details of source data and statistical analyses are provided in the Source Data file.

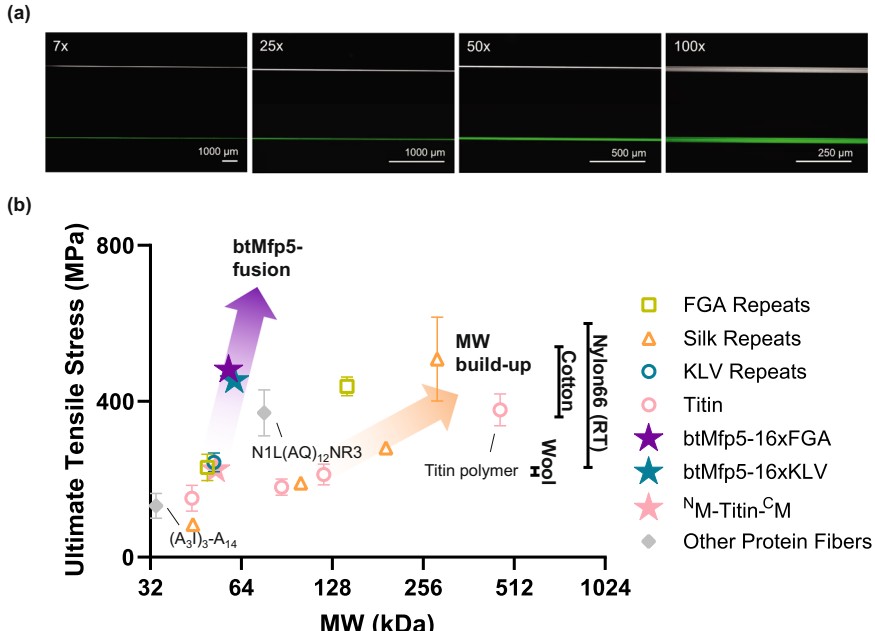

**Fig. 5 | btMfp5-fusion as an alternative strategy other than MW build-up to boost mechanical properties of protein-based fiber materials. a** Phase-contrast and fluorescent images of $^N$M-GFP-$^C$M fiber at different magnifications; **b** strength-MW plot of various protein-based fiber materials from this work and from previous works[9–11]. Data are presented as mean values ± standard deviations. Error bars represent standard deviation. Fiber average ultimate tensile strength, standard deviation and sample size data from previous works include FGA Repeats: 16xFGA, 48xFGA (left to right)[10]; Silk Repeats: 16-mer, 32-mer, 64-mer and 96-mer (left to right)[9]; KLV Repeats: 16xKLV[10]; Titin (water): Titin 4Ig, Titin 8Ig, Titin 12Ig and Titin polymer (left to right)[11]; Other Protein Fibers: (A3I)3-A14[44], N1L(AQ)12NR3[26]. Black lines on the right indicate typical values of strength of different conventional fibers (RT: regular tenacity) and are independent from the $x$-axis. The two arrows indicate the two strategies in designing mechanically-strong protein-based fiber materials. Source data are provided in the Source Data file.

simplifying protein synthesis, resulting in high protein titer ($8.0 \pm 0.70$ g/L). Our observation on the role of cation-π and π-π interaction are also consistent with these previous studies.

In summary, this work demonstrated that bi-terminal fusion of Mfp5 fragments boosted the mechanical properties of multiple recombinant protein fibers. Low MW proteins designed with this strategy (~60 kDa, e.g. $^N$M-16FGA-$^C$M) can be produced at high titers, and their fibers displayed impressively high strength and toughness. The combination of high protein yield and strong mechanical performance makes this engineered protein a promising target for industrial-scale manufacturing to enable practical applications. Our strategy highlights the advantage of self-interacting IDPs in promoting fiber mechanical properties. While btMfp5 fusion can be potentially used on a wide range of protein-based materials, this design principle opens opportunities to engineer other IDPs that better promote material properties, even beyond fiber materials.

## Methods

### Chemicals and reagents
All chemicals and reagents were procured from MilliporeSigma (Burlington, MA), unless specified otherwise. Plasmid cloning experiments were completed using gel extraction kits and plasmid miniprep kits

(iNtRON Biotechnology, Seoul, Republic of Korea) following the manufacturer's instructions. Phusion DNA polymerase, FastDigest restriction enzymes and T4 DNA ligase (Thermo Fisher Scientific, MA) were employed for all the PCR, digestions and ligations according to the manufacturer's recommended protocols. The Ni-NTA columns used in this study were purchased from (GE Healthcare, IL).

### Plasmid assembly
DNA sequences of 16x polymeric amyloid-silk and 16x recombinant spider silk repeats were obtained from plasmids pB6c-A-16xKLVFFAE-H10 and pE8a-A-16xFGAILSS-H10, respectively[10,51]. The titin fragment was originated from rabbit soleus titin domains Ig67-70 and was amplified from plasmid p-4XT[11,52]. The human Fyn tyrosine kinase SH3 domain was prepared using synthetic DNA (IDT Inc. Coralville, IA)[53]. The GFP sequence was obtained from a plasmid as used in a previous study[54]. The *Mytilus galloprovincialis* Mfp5 sequence was obtained from plasmid pE7a-mfp5$^{(1)}$ as described in recent studies[18,49]. Mfp5 mutants were prepared from synthetic DNAs (IDT Inc, Coralville, IA) and codon-optimized for *E. coli* expression. The N- and C-half Mfp5 coding sequences were incorporated between the Acc65I/NheI and BcuI/Kpn2I restriction sites of the pE8a plasmid. Primers used in this study were chemically synthesized (IDT Inc, Coralville, IA), and their

sequence information are provided in the associated supplementary data file. *E. coli* NEB10β strain (New England Biolabs, MA) was used for all plasmid construction and protein expression. *E. coli* cells were cultured in Luria Broth (LB) media (5 g/L yeast extract, 10 g/L tryptone and 10 g/L NaCl) with appropriate antibiotics of 50 µg/mL kanamycin or 100 µg/mL ampicillin on orbital shakers at 37 °C for plasmid cloning purposes.

## Protein expression in shake flasks

Protein expressions were performed in shake flasks using Terrific Broth (TB) media (24 g/L yeast extract, 20 g/L tryptone, 0.4% v/v glycerol, 17 mM $KH_2PO_4$ and 72 mM $K_2HPO_4$, pH = 7.2) containing 100 µg/mL ampicillin. Protein expression in shake flasks began with transformation of corresponding plasmids into *E. coli* NEB10β strain. Once colonies grew on LB agar (5 g/L yeast extract, 10 g/L tryptone, 10 g/L NaCl and 15 g/L agar) plates, a single colony was selected to inoculate a seed culture of 50 mL LB media and cultivated at 37 °C with orbital shaking until $OD_{600}$ reached 0.8. This seed culture was then used to inoculate TB media in 2 L shake flasks at 37 °C with orbital shaking until $OD_{600}$ reached 3-5. All cultures were induced by addition of 0.4% arabinose and were allowed to grow at 30 °C with orbital shaking for additional 24 h. Cells pellets were collected by centrifugation and were stored at −80 °C before proceeding to purification.

## Protein purification

Protein purification protocol was modified from previous methods[10,12,55]. For insoluble proteins, including all FGA and KLV proteins, with or without btMfp5 fusion, cell pellets were lysed in buffer A (6 M guanidine hydrochloride, 50 mM $K_2HPO_4$ and 300 mM NaCl, pH = 7.0) at a ratio of 40 mL buffer A per liter of cell culture, followed by centrifugation. The supernatant was subjected to Ni-NTA column chromatography and washed sequentially with buffer B (8 M urea, 50 mM $K_2HPO_4$ and 300 mM NaCl, pH = 7.0) containing 0 mM, 20 mM, and 50 mM imidazole. The target proteins were eluted using buffer B containing 300 mM imidazole. For water-soluble proteins, such as titin, GFP, and SH3, cell pellets were lysed in PBS buffer (137 mM NaCl, 2.7 mM KCl, 1.8 mM $KH_2PO_4$ and 10 mM $Na_2HPO_4$, pH = 7.4) at a ratio of 40 mL PBS buffer per liter of cell culture, using sonication followed by centrifugation. The supernatant was loaded onto a Ni-NTA column, washed with PBS buffer containing 0 mM, 20 mM, and 50 mM imidazole, and eluted with PBS buffer containing 300 mM imidazole. The purified proteins were dialyzed against 5% acetic acid, lyophilized, and stored at −80 °C before dissolved in HFIP to make spinning dopes.

## SDS-PAGE

All SDS-PAGE gels were 1 mm thick and discontinuous with 5% stacking gel on the top and indicated percentages separation gels on the bottom. Samples were resuspended in 1x Laemmli sample buffer (10% v/v glycerol, 2% w/v SDS, 60 mM Tris-HCl, 0.01% w/v bromophenol blue and 100 µM DTT, pH = 6.8) and incubated at 95-100 °C for 10 min. Electrophoresis were performed on Mini-PROTEAN Tetra Cells (Bio-Rad, CA) in 1x Tris-glycine SDS buffer (25 mM Tris-HCl, 250 mM glycine and 0.1% w/v SDS, pH = 8.8). For staining, all SDS-PAGE gels were incubated in a staining buffer (40% v/v methanol, 10% v/v acetic acid, 0.1% w/v Coomassie Brilliant Blue G-250 in water) for at least 1 h, before transferred to a destaining buffer (40% v/v methanol, 7% v/v acetic acid in water) and destained on orbital shakers at room temperature. Gel images were acquired using Azure Biosystems cSeries (Azure Biosystems Inc., CA) and were analyzed by ImageJ 1.53 software[56].

## Bioproduction in fed-batch bioreactors

Fed-batch fermentation of $^NM$-16xFGA-$^CM$ from pET28a vector in *E. coli* BL21(DE3) strain (New England Biolabs, MA) was performed following published protocols[41,57]. Specifically, cells were cultivated in 600 mL TB media with 50 µg/mL kanamycin until $OD_{600}$ reached ~5. The culture was then centrifuged, and the pellet was resuspended in 1 L batch media (10 g/L glucose, 60 g/L glycerol, 20 g/L tryptone, 24 g/L yeast extract, 0.5 g/L magnesium sulfate heptahydrate, 3.4 g/L potassium phosphate monobasic, 3.6 g/L sodium phosphate dibasic, 2.7 g/L ammonium chloride, 0.7 g/L sodium sulfate, 100.8 mg/L ferric citrate, 2.5 mg/L cobalt chloride hexahydrate, 15 mg/L manganese chloride tetrahydrate, 1.5 mg/L copper chloride dihydrate, 3 mg/L boric acid, 2.1 mg/L sodium molybdate dihydrate, 33.8 mg/L zinc acetate hydrate, 14.1 mg/L EDTA, 50 mg/L kanamycin, 0.01% v/v antifoam 204), and transferred to a 2 L Bioflo120 fed-batch bioreactor (Eppendorf, Hamburg, Germany). The culturing was controlled at 29 °C, and the pH was maintained at 7.1 with either 3 M phosphoric acid or 25% ammonia. Autoclaved water was supplemented to the system in compensation for water losses during fermentation. The DO level was initially set at 70% and maintained at no less than 30% by adjusting the stirring speed (200-1200 rpm) and the airflow rate. Cells were induced at $OD_{600} \approx 68$ with a final IPTG concentration of 0.25 mM. Feed media (400 g/L glycerol, 20 g/L tryptone, 24 g/L yeast extract, 40 g/L magnesium sulfate heptahydrate, 3.4 g/L potassium phosphate monobasic, 3.6 g/L sodium phosphate dibasic, 2.7 g/L ammonium chloride, 0.7 g/L sodium sulfate, 40 mg/L ferric citrate, 4 mg/L cobalt chloride hexahydrate, 23.5 mg/L manganese chloride tetrahydrate, 2.3 mg/L copper chloride dihydrate, 4.7 mg/L boric acid, 4 mg/L sodium molybdate dihydrate, 16 mg/L zinc acetate hydrate, 13 mg/L EDTA, 50 mg/L kanamycin, 0.01% v/v antifoam 204) was added at a rate of 0.2 mL/min. Pellets were collected 20 h after induction. Estimation of protein expression level and titer was completed following the protocol in Supplementary Notes.

## Fiber spinning

Fiber spinning protocol was adopted from previous publications with some modifications[9,10,12]. Spinning dopes were prepared by the dissolution of lyophilized protein powders in HFIP. The resulting solution was loaded to a 100 µL Hamilton syringe (Hamilton Robotics, NV), and extruded into a 95% v/v methanol bath (pH = 8.0) using a Harvard Apparatus Pump 11 Elite syringe pump (Harvard Apparatus, MA) at a controlled protrusion rate of 10 µL/min. To achieve pH-controlled spinning, either 5 mM $CH_3COONa$ or 5 mM $Na_2CO_3$ was added to the 95% v/v methanol bath to adjust the pH to 5.5 or 11, respectively. The extruded fibers were then transferred to a 75-80% v/v methanol bath and extended to five times their original length. Following extension, the fibers were removed from the methanol bath and allowed to dry in a ventilated, dark, and dry room at room temperature until further testing.

## Mechanical testing

Tensile tests were conducted on an MTS Criterion Model 41 universal test frame equipped with a 1 N load cell (MTS Systems Corporation, MN). The tests were carried out under controlled environmental conditions of 20% relative humidity (RH) and 25 °C temperature, and with a constant pulling speed of 10 mm/min. Stress-strain curves were obtained at a 50 Hz sampling rate. To calculate the mechanical properties, engineer stress and strain were employed. Specifically, the ultimate tensile strength was calculated as the maximum load divided by the initial fiber cross-sectional area, which was determined by light microscopy before testing, assuming a circular cross-section. The modulus was obtained by fitting a linear least-square regression line to the initial elastic region of the stress-strain curve. The ultimate breaking strain was calculated as the percentage of fiber elongation relative to the initial gauge length of 5 mm before fracture. The toughness was determined as the area under the total stress-strain curve. The MTS TestSuite TW Elite software (MTS Systems Corporation, MN) was used for data collection and mechanical properties calculation.

## Light microscopy

Fiber diameters were measured using a Zeiss Axio Observer ZI inverted microscope (Zeiss, Stuttgart, Germany) with a phase-contrast 20x objective lens. The quantification was completed using the Axiovision LE software (Zeiss, Stuttgart, Germany).

## SEM

Fibers samples were acquired after tensile tests and were mounted onto a sample holder using conductive tapes. The sample holder was sputtered with 10 nm gold using a Leica EM ACE600 high-vacuum sputter coater (Leica Microsystems, Wetzlar, Germany). Imaging was completed with a Thermofisher Quattro S Environmental Scanning Electron Microscope (Thermofisher Scientific, MA) at an accelerating voltage of 10 kV and a spot size of 3.0, and images were acquired using AutoScript 4 software (Thermofisher Scientific, MA).

## Fluorescence microscopy

Phase-contrast images of the $^{N}M$-GFP-$^{C}M$ fiber were observed by the fluorescence stereomicroscope (ZEISS Axio Zoom V16) equipped with a 7x, 25x, 50, and 100x lens. For the fluorescence images, the 7x lens required 550 msec exposure time, others were set as 280 msec. Tri-channel (EGFP, DAPI, DsRed) fluorescent images were captured at the same magnification (80x), with exposure time of 90 ms, using Hamamatsu ORCA-Flash4.0 V3 sCMOS camera equipped in Zeiss Axio Zoom V16 in 16 bit and processed using Zen Software (Zeiss, Stuttgart, Germany).

## Polarized Raman spectroscopy analysis

Polarized Raman Spectroscopy were recorded following published methods[12,58–60]. Specifically fiber samples were fixed by tape on a glass slide. Raman spectra were acquired with a Renishaw RM1000 InVia confocal Raman spectrometer (Renishaw, Wotton-under-Edge, United Kingdom) coupled to a Leica DM LM microscope with rotating stage (Leica Microsystems, Wetzlar, Germany). Fibers were initially oriented along the x-axis and were irradiated using a 514 nm line argon laser with polarization fixed along the x-axis and focused through a 50x objective (NA = 0.75). Fibers were then aligned to the y-axis, followed by the same data acquisition process. Spectra were recorded from 1150 to 1750 $cm^{-1}$ with an 1800 lines/mm grating. For each acquisition, a total of 16 spectra were accumulated, each for 10 s. All fibers remained intact after acquisition with no visual sign of degradation under the incident laser. Spectra collected were analyzed with Fityk 1.3.1[61]. Baseline subtraction is accomplished using a built-in Fityk automatic convex hull algorithm. All spectra were normalized to the intensity of the 1447 $cm^{-1}$ peaks, which arises from $CH_3$ asymmetric stretching and $CH_2$ bending[59] and is insensitive to protein conformation.

## Statistics and reproducibility

Statistical analyses, including student t-test and ANOVA, were conducted with Microsoft Excel 365 (Microsoft Corporation, WA) and GraphPad Prism 9 (GraphPad Software, CA). All data are presented means ± SD as depicted in the figure caption. Significant statistical difference is considered at $p < 0.05$ and is denoted with asterisks (*), while no significant statistical difference is considered at $p \geq 0.05$ and is denoted with 'ns' in figures. Statistical tests for significance were described in individual figure legends. Details about statistical analyses can also be found in the Source Data file. Details about statistical analyses in Fig. 5b can be found in the corresponding references if those results were not generated in this study. No data were excluded from the analyses. For representative experiments (imaging), each experiment has been repeated for at least three times independently with similar results.

## Reporting summary

Further information on research design is available in the Nature Portfolio Reporting Summary linked to this article.

## Data availability

All experimental data from mechanical testing, Raman spectra, uncropped raw gel images, protein sequences and oligonucleotide sequences generated in this study underlying all main figures and supplementary figures are provided as a Source Data file. All protein sequences in this study have been provided in Supplementary Dataset 1. Protein structures with accession numbers "5E61", "3OW9", "3B43", "1EMB" and "3UA6" were used in this work and obtained from the RCSB Protein Data Bank. All remaining data generated, processed, or analyzed in this study are available within the article and Supplementary Information or from the corresponding author upon reasonable request. Source data are provided with this paper.

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

## Acknowledgements

The authors would like to thank Dr. Srikanth Singamaneni and Yixuan Wang for their assistance with the Raman spectrometry. SEM images were acquired within the facilities of IMSE at Washington University. This research was funded by United States Department of Agriculture (grant number 20196702129943 to F.Z.) and National Science Foundation (award numbers DMR-2207879 and OIA-2219142 to F.Z.).

## Author contributions

F.Z. and J.L. conceived the project. J.L. performed plasmid construction, strain engineering, cell culture, protein expression, protein purification, SDS-PAGE, fiber spinning, optical microscopy, SEM imaging, fiber

mechanical testing, Raman analyses and associated analyses. B.J. performed fluorescence imaging. X.C. and H.Y. performed protein expression and protein purification. Y.H. helped with the plasmid design. J.L. and F.Z. prepared the manuscript with comments from all authors.

## Competing interests

The authors declare the following competing financial interest(s): J.L. and F.Z. have filed a provisional patent application based on the protein sequences and methods of fabrication presented in this work (US provisional application No. 63/488,637, filed Mar. 6th, 2023). The remaining authors declare no competing interests.
