## [Peer Review File · Nature Communications]

REVIEWER COMMENTS

Reviewer #1 (Remarks to the Author):

Summary:

The authors engineer split Mfp5 (sMfp5) protein as a strategy to improve upon challenges present in the field of microbially-synthesized materials. The cation- π stacking, and π - π stacking interactions of the engineered splits allow for making mechanically strong recombinant fibers from low-molecular weight proteins opening up new opportunities for high yield manufacturing of microbially-synthesized protein-based materials. They have implemented their system to make fiber materials from different proteins such as SH3 and GFP which shows applicability of their system. Overall the research is of high quality and well written, and seems to move the field of synthetic biology one step forward.

Impact:

The general applicability of engineering split Mfp5 protein for making fibrous materials from different proteins makes the work high impact in the field. Overall, the logic behind experiments and data analysis make sense. While I think the work is important, several issues lessened my enthusiasm as described below.

Comments/Concerns

1. In the abstract please specify the molecular mechanism of split sMfp5 proteins so reader can follow. What is the underlying mechanism that they form a fiber. Authors have this covered in the intro but it worth incorporating in the abstract for flow.
2. To enhance readability please define acronyms before use. For example, SH3 lane 40, DOPA lane 77,
3. 2g- For more clear comparison, perhaps present the data graph as mean of replicates including the error bars and present current 2g in supplementary material.
4. For protein expression in shake flasks, can you clarify why you would grow culture to such high OD (3-5) and why you would incubate for additional 24 hours post induction?
5. Please clarify on the methods section why pellets were lysed over 12 hours of stirring vs other methods such as sonication/homogenization.
6. Supplementary Figure S15-b, authors estimate protein yield of 8.0g/L using a qualitative method by comparing band sizes on the gel. Since yield is one of the essential advantages of their system, the authors need a more quantitative method to present their data. I understand use of gel for yield estimation is to estimate total protein in lysate which is impossible to quantify using BSA or Nanodrop. However, since they have the purified protein, they can report quantity by either nanodrop or Bradford assay or at least have replicates on the gel to eliminate human error associated with loading gels. In addition, more clarification is needed in the method section on how and why they estimated yield using this method. They emphasize on this number in both their abstract and discussion so worth taking the time to provide solid data.
7. Authors mention that intermolecular interactions among split Mfp5 is pH dependent. In order for this work to be widely applicable to the field and their systems being incorporated into other's research more readily, the authors need to assess fiber formation and mechanical strength over range of pH. In addition, authors need to clarify pH for all their mechanical strength tests throughout the manuscript.

8. Showing microscopy data, please provide at least one example with multiple magnification panels of the same sample. For easy visualization, I recommend the NM-GFP-CM. By seeing the data at lower magnification then comment on percentage of the sample that formed fiber complexes as expected and discuss efficiency of your developed system.

9. Please expand the discussion on the impact of the work

Reviewer #2 (Remarks to the Author):

1. What are the noteworthy results?

-Demonstration of a versatile strategy for developing high yield-small protein-based fibers with good mechanical strength and interesting properties provided from fused functional domain.

-Utilization of newly engineered split Mfp5 and investigation in its principles in enhancing protein fiber mechanical strength.

2. Will the work be of significance to the field and related fields? How does it compare to the established literature? If the work is not original, please provide relevant references.

It is foreseen that this strategy will be utilized in producing silk materials with high yield and intrinsic biological functionalities. Afterall, the low production yield of protein fiber is a main obstacle for actual applications of protein fiber materials in industries.

Quality is sufficient as compared to other N.C. articles.

3. Does the work support the conclusions and claims, or is additional evidence needed?

Major: To justify the necessity of splitting Mfp5 to promote head-to-tail intermolecular interactions, the control M-16xFGA-M is lacking. It is already stated in the article that Mfp5 can already self-interact and using 16xFGA-M as justification is unconvincing.

Besides, there is no direct evidence that the split-Mfp5 pair can form specific head-to-tail interactions. While the split fragments derived from globular proteins such as split GFP can reconstitute spontaneously or upon induction, it is just unfathomable for the sMfp5 fragments derived from an intrinsically disordered protein to reassemble specifically. In other words, unless proven otherwise, NM-16xFGA-CM will be no different from NM-16xFGA-NM or CM-16xFGA-CM or full length M-16xFGA-M.

Minor: In Figure 5b, it is suggested to add a right y-axis labeled as yield to relate the yield with MW and ultimate tensile strength. Also, it would be nice if comparisons with existing silks from other people's studies could be made.

4. Are there any flaws in the data analysis, interpretation and conclusions? - Do these prohibit publication or require revision?

No major problems.

Minor problems:

a. Line 41: Add back "(IDPs)" following "intrinsically-disordered proteins" as the abbreviation is not found in the body.

b. Line 68-70: Add references for "Metabolic and genetic engineering methods are useful in enhancing protein titers and yields;15 however, these strategies have limited effects 70 on high MW PBM".

c. Line 261-263: Add references 14, 49 for "Mechanical enhancement on recombinant silk fibers was observed when globular terminal domains of spidroins were fused to a minimal repetitive domain (2 repeats).", same as that in line 377.

d. Line 288-291: Confusing description. It most probably means making the protein construct 16xFGA-M but can also mean adding CMfp5 after making 16xFGA-NMfp5 and let them assemble into 16xFGA-M. In the latter case, affinity of split Mfp5 assembly has to be tested (e.g. ITC).

e. Line 292: Add "tensile" in-between ultimate strength.

f. Figure 4: Switch Figure 4m-p with Figure 4q-t to corresponds to the sequence as presented in Table 12-15 and Line 337-339.

g. Figure S15: Make the lane number of the SDS-PAGE corresponds to the description.

h. Line 365-368: Add references for "While these large proteins can eventually generate strong fibers,

their yields were no match to low MW proteins as demonstrated in this study, and they often require complicated purification processes such as ammonia sulfate precipitation and multi-step chromatography (e.g. size exclusion chromatography).”.

5. Is the methodology sound? Does the work meet the expected standards in your field?

Yes.

6. Is there enough detail provided in the methods for the work to be reproduced?

Yes.

Dear reviewers,

Please note that the line numbers in this response corresponds to the marked copy of the revised manuscript.

Reviewer 1:

1. In the abstract please specify the molecular mechanism of split sMfp5 proteins so reader can follow. What is the underlying mechanism that they form a fiber. Authors have this covered in the intro but it worth incorporating in the abstract for flow.

Response: We thank the reviewer for the suggestion. We have clarified the mechanism in the abstract in lines 26, 29-32.

2. To enhance readability please define acronyms before use. For example, SH3 lane 40, DOPA lane 77,

Response: We appreciate the reviewer for pointing this out. We have defined these acronyms in line 33-35 for SH3, and line 70 for DOPA.

3. 2g- For more clear comparison, perhaps present the data graph as mean of replicates including the error bars and present current 2g in supplementary material.

Response: The means of replicates and error bars of mechanical data are presented in Fig. 2d and Table S3. To promote clarity, we reorganized Figs 2g to present 3 representative stress-strain curves. All other stress-strain curves are presented in supplementary information Figure S2. The raw data of all measurements are provided in the source data file.

4. For protein expression in shake flasks, can you clarify why you would grow culture to such high OD (3-5) and why you would incubate for additional 24 hours post induction?

Response: We appreciate this question. We understand that common recombinant proteins are usually induced at a relatively low OD between 0.6 and 0.8 when cells are in log phase and incubate for 4-6 hours. For expression of material proteins, we found that induction at high-OD gave better results, and this method has been used in previous papers for expression of similar proteins (10.1021/acssynbio.9b00362, 10.1021/acs.biomac.9b00218, 10.1021/acsnano.1c02944). We believe these conditions work better because expressing these material proteins are

burdensome to host cells. After induction, cell growth is very slow, thus induction at a higher OD provides higher protein yields. Additionally, these proteins do not need to fold upon expression and are stable in microbial hosts. So 24 hours of expression gave us higher protein yields than 6-hour expression (see Figure R1).

Figure R1. Expression of ^NM-16xFGA-^CM. Lane 1, MW marker; Lane 2, *E. coli* whole cell lysate before induction; Lane 3, *E. coli* whole cell lysate after 6-hour induction; Lane 4, *E. coli* whole cell lysate after 24-hour induction.

5. Please clarify on the methods section why pellets were lysed over 12 hours of stirring vs other methods such as sonication/homogenization.

Response: We use different methods to lyse cells depending on the solubility of the target proteins. For insoluble proteins, such as ^NM-16xFGA-^CM, 6 M guanidine hydrochloride was added to cell pellet and stirred for 12 hours. This method allowed us to lyse cells and extract target proteins in one step and has been previously used for similar proteins (10.1021/acs.biomac.9b00218, 10.1021/acsnano.1c02944). For relatively soluble proteins (i.e. ^NM-Titin-^CM and ^NM-GFP-^CM), we lysed cells by sonication in PBS buffer. These proteins were then dissolved in aqueous solution and can be separated from cell debris. We added these details to the Methods section in Line 352-354 to be clearer.

6. Supplementary Figure S15-b, authors estimate protein yield of 8.0g/L using a qualitative method by comparing band sizes on the gel. Since yield is one of the essential advantages of their system, the authors need a more quantitative method to present their data. I understand use of gel for yield estimation is to estimate total protein in lysate which is impossible to quantify using BSA or Nanodrop. However, since they have the purified protein, they can report quantity by either nanodrop or Bradford assay or at least have replicates on the gel to eliminate human error associated with loading gels. In addition, more clarification is needed in the method section on how and why they estimated yield using this method. They emphasize on this number in both their abstract and discussion so worth taking the time to provide solid data.

Response: We appreciate the question raised by the reviewer. We followed previously published methods (10.1016/j.mattod.2021.07.020, 10.1002/adfm.202200986) to quantify protein expression titer and yield from bioreactors. To further assure the reported protein yield, we followed the reviewer's recommendation by replicating the SDS-PAGE three times to eliminate error associated with loading gels. The new gels and standard curves were added (Figure R2-R3). We found that protein quantities between different gels are highly consistent. The estimated expression levels and titers from the second gel are 12.8% and 7.3 g/L (Figure R2), and from the third gel are 13.1% and 8.7 g/L (Figure R3), respectively. Using values from three separate gels, we calculated mean and standard deviation of expression level to be $13.5 \pm 1.0\%$ of total *E. coli* proteome, and titer to be 8.0 ± 0.70 g/L. These additional quantification results are provided in revised Supplementary Figure S19-S21 and in lines 29, 85, 245-246 of the revised manuscript. Additional clarification on the quantification method was provided to the Supplementary note at the end of Supplementary Materials.

Figure R2. Replicate experiment of SDS-PAGE and ¹⁵N-16xFGA-¹³C protein quantification produced from a 2 L fed-batch bioreactor. Lane 1, MW marker; Lane 2, *E. coli* whole cell lysate before induction; Lane 3, *E. coli* whole cell lysate after 20h induction; Lane 4-7, purified ¹⁵N-16xFGA-¹³C(KRtoS) protein samples with concentrations of 0.0625, 0.125, 0.25 and 0.5 mg/mL. Concentration-band area linear

dependency from purified ^{15}N -16xFGA- ^{13}C M(YtoS) bands. The green cross represents the band area of the ^{15}N -16xFGA- ^{13}C M band on lane 3 in gel image.

Figure R3. Triplicate experiment of SDS-PAGE and ^{15}N -16xFGA- ^{13}C M protein quantification produced from a 2 L fed-batch bioreactor. Lane 1, MW marker; Lane 2, *E. coli* whole cell lysate before induction; Lane 3, *E. coli* whole cell lysate after 20h induction; Lane 4-8, purified ^{15}N -16xFGA- ^{13}C M(KRtoS) protein samples with concentrations of 0.0625, 0.125, 0.25, 0.5 and 1 mg/mL. Concentration-band area linear dependency from purified ^{15}N -16xFGA- ^{13}C M(KRtoS) bands. The green cross represents the band area of the ^{15}N -16xFGA- ^{13}C M band on lane 3 in gel image.

7. Authors mention that intermolecular interactions among split Mfp5 is pH dependent. In order for this work to be widely applicable to the field and their systems being incorporated into other's research more readily, the authors need to assess fiber formation and mechanical strength over range of pH. In addition, authors need to clarify pH for all their mechanical strength tests throughout the manuscript.

Response: We thank the reviewer for raising the issue of pH. We expect that intermolecular interactions between sMfp5 to be pH dependent because as pH decreases sMfp5 is expected to be more strongly positively charged, thus promoting cation- π interactions and fiber ultimate strength. To test this hypothesis, we followed the reviewer's suggestion by spun the ^{15}N -16xFGA- ^{13}C M(YtoS) protein in a range of pH from 5.5 to 11 and tested fiber mechanical properties. We chose the ^{15}N -16xFGA- ^{13}C M(YtoS) protein because its ^{13}C M lacks tyrosine, thus reducing the influence from π - π interaction. We varied the pH by changing the pH of coagulation bath during fiber spinning. Comparing both fiber ultimate strength and toughness, we indeed observed enhanced mechanical properties as pH increased. When fibers were spun at pH=5.5, ultimate tensile stress was 18% and 31% higher than those at pH 8.0 and pH 11, while toughness was 48% and 67% higher than those at pH 8.0 and pH 11, respectively (Figure R4). We thank

the reviewer's suggestion that allowed us to find spinning conditions to produce even stronger fibers (at pH 5.5, ultimate strength 481 ± 31 MPa and toughness 179 ± 39 MJ*m⁻³). We have presented these new results in the revised manuscript. Please find the changes in line 184-197 and Figure 3g-i, Figure S7, and Table S9 and S10.

We further clarified the pH for all other experiments. In general, the wet-spinning protocol used in this manuscript followed previous papers (10.1073/pnas.1003366107, 10.1021/acs.biomac.8b00980). The coagulation bath contained 95% methanol and 5% water. Its pH, as measured by pH meter, is ~8.0.

Figure R4. Tensile test results of ^NM-16xFGA-^CM(YtoS) spun at different pH. (a) Summary of ultimate tensile stress for ^NM-16xFGA-^CM(YtoS) fibers spun at different pH. (b) Summary of toughness for ^NM-16xFGA-^CM(YtoS) fibers spun at different pH. (c) Representative stress-strain curves of ^NM-16xFGA-^CM(YtoS) fibers spun at different pH. Error bars represent standard deviation. ***p < 0.001, ****p < 0.0001, two-tailed unpaired t test. For all mechanical tests, n = 10.

8. Showing microscopy data, please provide at least one example with multiple magnification panels of the same sample. For easy visualization, I recommend the NM-GFP-CM. By seeing the data at lower magnification then comment on percentage of the sample that formed fiber complexes as expected and discuss efficiency of your developed system.

Response: We thank the reviewer for the suggestion. Please see Figure R5 containing multiple magnification panels of the ^NM-GFP-^CM fiber. These figures were also added to Figure 5a. Because the ^NM-GFP-^CM fibers are fluorescent, we conclude that at least some of the proteins folded into the beta-barrel structure as the wild-type GFP in aqueous solution. It is rather difficult to determine the percentage of the ^NM-GFP-^CM protein that form complexes, as that require molecular resolution of the fiber materials.

Figure R5. Phase-contrast and fluorescent images of a ^NM-GFP-^CM fiber at different magnifications.

9. Please expand the discussion on the impact of the work

Response: We thank the reviewer for the suggestion. We have expanded the discussion on the impact of the work. Please see our changes in Line 305-310.

Reviewer 2:

1. To justify the necessity of splitting Mfp5 to promote head-to-tail intermolecular interactions, the control M-16xFGA-M is lacking. It is already stated in the article that Mfp5 can already self-interact and using 16xFGA-M as justification is unconvincing.

Besides, there is no direct evidence that the split-Mfp5 pair can form specific head-to-tail interactions. While the split fragments derived from globular proteins such as split GFP can reconstitute spontaneously or upon induction, it is just unfathomable for the sMfp5 fragments derived from an intrinsically disordered protein to reassemble specifically. In other words, unless proven otherwise, NM-16xFGA-CM will be no different from NM-16xFGA-NM or CM-16xFGA-CM or full length M-16xFGA-M.

Response: We appreciate the reviewer for raising this point and apologize for not making it clear in our original manuscript. We agree with the reviewer that interactions between intrinsically disordered sMfp5 fragments are not limited to any specific form/structure of interactions, so the mechanism is different from split GFP, split luciferase, and split dihydrofolate reductase, all of which have specific interactions/structures. We used the term “head-to-tail interactions” in our original manuscript to describe the way of interaction between ^NM (head of the ^NM-16xFGA-^CM

protein) and ^CM (tail of the ^NM-16xFGA-^CM protein) in our fibers. The control protein 16xFGA-M does not have this “head-to-tail interactions” between protein chains, thus its fiber does not display enhanced mechanical properties. We further showed that the “head-to-tail interactions” between sMfp5 fragments mostly involve in general cation- π and π - π interactions, thus are not limited to interactions between any specific set of residue pairs (e.g. Y11 with K606).

Because each sMfp5 fragment contains both tyrosine and positively-charged residues, we also agree with the reviewer that ^NM can interact with ^NM, and ^CM can interact with ^CM. Thus we expect both ^NM-16xFGA-^NM and ^CM-16xFGA-^CM fibers have similar “head-to-tail interactions” (head of a protein interacts with the tail of another protein, in these cases “head” and “tail” share the same sequence), leading to enhanced mechanical properties. To prove this, we cloned and tried to express ^NM-16xFGA-^NM, ^CM-16xFGA-^CM, M-16xFGA-M proteins. We further prepared ^CM-16xFGA-^CM fibers and measured their mechanical properties following the same procedures as other fibers. Indeed, ^CM-16xFGA-^CM fibers exhibited enhanced mechanical properties compared to those of 16xFGA fibers. Both tensile strength and toughness of ^CM-16xFGA-^CM fibers are similar to those of ^NM-16xFGA-^CM fibers (Figure R6). Unfortunately, ^NM-16xFGA-^NM and M-16xFGA-M proteins were not expressed due to genetic instability (recombination between identical sMfp5 DNA sequences). But we believe the ^CM-16xFGA-^CM fiber results have proved the point. We have added these results to line 198-207 in the revised manuscript and Supplementary Figures S8-S9. We further clarified our manuscript by changing the term “head-to-tail” to “end-to-end” in the revised manuscript in Line 26, 75.

Figure R6. Tensile test results of ^CM-16xFGA-^CM fibers. (a) Stress-strain curves of ^CM-16xFGA-^CM fibers; (b) Summary of ultimate tensile stress of 16xFGA, ^NM-16xFGA-^CM and ^CM-16xFGA-^CM fibers. (c) Summary of toughness for 16xFGA, ^NM-16xFGA-^CM and ^CM-16xFGA-^CM fibers. Error bars represent standard deviation. **p < 0.01, ****p < 0.0001, two-tailed unpaired t test. For 16xFGA and ^NM-16xFGA-^CM, n = 10; for ^CM-16xFGA-^CM, n = 8.

2. In Figure 5b, it is suggested to add a right y-axis labeled as yield to relate the yield with MW and ultimate tensile strength. Also, it would be nice if comparisons with existing silks from other people's studies could be made.

Response: We thank the reviewer for the suggestion. We have revised Figure 5b by including values of existing silks from other people's studies. Regarding protein yields, we found adding yields to Figure 5b makes it too crowded, we thus included another plot to compare protein titers with past studies. The results are presented in Supplementary Table S19 and Figure S22. Please note that many past studies did not report protein yields, so the number of data points in Figure S22 is significantly less than in Figure 5b.

3. Line 41: Add back "(IDPs)" following "intrinsically-disordered proteins" as the abbreviation is not found in the body.

Response: We have made the change in line 35.

4. Line 68-70: Add references for "Metabolic and genetic engineering methods are useful in enhancing protein titers and yields;15 however, these strategies have limited effects 70 on high MW PBM".

Response: We thank the reviewer for raising this issue. We have added reference to this claim. Please see changes we made in Line 60.

5. Line 261-263: Add references 14, 49 for "Mechanical enhancement on recombinant silk fibers was observed when globular terminal domains of spidroins were fused to a minimal repetitive domain (2 repeats).", same as that in line 377.

Response: We thank the reviewer for pointing out our omission. We have added these references to the claim. Please see the change in line 125.

6. Line 288-291: Confusing description. It most probably means making the protein construct 16xFGA-M but can also mean adding CMfp5 after making 16xFGA-NMfp5 and let them assemble into 16xFGA-M. In the latter case, affinity of split Mfp5 assembly has to be tested (e.g. ITC).

Response: We thank the reviewer for raising this issue. We intended to say that we made the protein construct 16xFGA-M. We have revised the description to avoid confusion. Please see line 154-157.

7. Line 292: Add “tensile” in-between ultimate strength.

Response: We have made this correction throughout the manuscript. Please see Lines 168, 182.

8. Figure 4: Switch Figure 4m-p with Figure 4q-t to corresponds to the sequence as presented in Table 12-15 and Line 337-339.

Response: We thank the reviewer for the suggestion. Rather than changing the figures, we changed the order of description in the manuscript to achieve the same goal. Please see our changes in Line 234-237.

9. Figure S15: Make the lane number of the SDS-PAGE corresponds to the description.

Response: We have revised the figure (now Fig S19) to corresponds to the description.

10. Line 365-368: Add references for “While these large proteins can eventually generate strong fibers, their yields were no match to low MW proteins as demonstrated in this study, and they often require complicated purification processes such as ammonia sulfate precipitation and multi-step chromatography (e.g. size exclusion chromatography).”.

Response: We thank the reviewer for raising this issue. We have added reference for this claim. Please see our changes in Line 261.

REVIEWER COMMENTS

Reviewer #1 (Remarks to the Author):

I read their response and they have addressed all the concerns that were mentioned either in text or with experiments. I am especially glad that the pH screen experiment helped them to find even a stronger fiber and improve upon their results.

Reviewer #2 (Remarks to the Author):

According to the new data, CM-16xFGA-CM, with the C-terminal of Mfp only, can form fibres comparable to NM-16xFGA-CM. Therefore the major conclusion of the paper, centered around split Mfp, needs to be redrawn and revised to reflect this new finding. It seems a truncated version of Mfp, instead of split Mfp, that is more essential for the system to work. The other technical revisions are good.

REVIEWER COMMENTS

Reviewer #2 (Remarks to the Author):

1. According to the new data, CM-16xFGA-CM, with the C-terminal of Mfp only, can form fibres comparable to NM-16xFGA-CM. Therefore the major conclusion of the paper, centered around split Mfp, needs to be redrawn and revised to reflect this new finding. It seems a truncated version of Mfp, instead of split Mfp, that is more essential for the system to work. The other technical revisions are good.

Response: We thank the reviewer for pointing out this issue. To address this issue, we have changed the title of the manuscript to “Bi-terminal Fusion of Intrinsically Disordered Mussel Foot Protein Fragments Boosts Mechanical Strength for a Wide Range of Protein Fibers”. We have also revised descriptions in the main text extensively to avoid confusion. Please see our changes in lines marked in blue.